

# Semi or fully automatic tooth segmentation in CBCT images: a review

Qianhan Zheng[1], Yu Gao[1], Mengqi Zhou[1], Huimin Li[1], Jiaqi Lin[1], Weifang Zhang[1,2] and Xuepeng Chen[1,3]

[1] Stomatology Hospital, Zhejiang University School of Medicine, Hangzhou, China
[2] Social Medicine & Health Affairs Administration, Zhejiang University, Hangzhou, China
[3] Clinical Research Center for Oral Diseases of Zhejiang Province, Cancer Center of Zhejiang University, Hangzhou, China

## ABSTRACT

Cone beam computed tomography (CBCT) is widely employed in modern dentistry, and tooth segmentation constitutes an integral part of the digital workflow based on these imaging data. Previous methodologies rely heavily on manual segmentation and are time-consuming and labor-intensive in clinical practice. Recently, with advancements in computer vision technology, scholars have conducted in-depth research, proposing various fast and accurate tooth segmentation methods. In this review, we review 55 articles in this field and discuss the effectiveness, advantages, and disadvantages of each approach. In addition to simple classification and discussion, this review aims to reveal how tooth segmentation methods can be improved by the application and refinement of existing image segmentation algorithms to solve problems such as irregular morphology and fuzzy boundaries of teeth. It is assumed that with the optimization of these methods, manual operation will be reduced, and greater accuracy and robustness in tooth segmentation will be achieved. Finally, we highlight the challenges that still exist in this field and provide prospects for future directions.

# INTRODUCTION

Radiographic images are indispensable for diagnosing and designing treatments in the medical field. X-ray, the most commonly used radiological examination method, has difficulty detecting lesions due to the overlapping tissues caused by the absorption of certain organs in the human body (*Oprea et al., 2008*). Computed tomography (CT) was developed to overcome these limitations; this method involves conducting cross-sectional scans around specific body parts one by one to obtain tomographic images (*Goldman, 2007*), compensating for the drawbacks of X-rays. Starting in the second half of the 1990s (*Loubele et al., 2006*), specialized dental cone beam computed tomography (CBCT) was widely used in various fields, including orthodontics (*Kapila & Nervina, 2015*), periodontology (*Woelber et al., 2018*), implant dentistry (*Jacobs et al., 2018*), temporomandibular joint diseases (*Larheim et al., 2015*), and endodontics (*Patel et al., 2019*). As documented by *Pauwels et al. (2000)*, it provide high-resolution images with less radiation exposure than traditional CT.

Corresponding authors
Weifang Zhang, chzwf@zju.edu.cn
Xuepeng Chen, cxp1979@zju.edu.cn

The progress and widespread utilization of CBCT have profoundly advanced the digitization of dental imaging, constituting a crucial component of the contemporary workflow of digital dentistry (*Eaton, 2022*). Tooth segmentation *via* CBCT primarily involves discerning distinctive density features (*Pal & Pal, 1993*) in imaging to distinguish teeth from the periodontal ligament and alveolar bones. This process ensures the precise extraction of each tooth from the CBCT image, allowing for a meticulous analysis of the morphology, position, and relationships of the tooth with surrounding structures. It enables the simulation and evaluation of diverse treatment modalities in a virtual environment, thereby significantly influencing how dental professionals approach diagnostics, treatment planning, and patient care. In orthodontic analysis, three-dimensional (3D) tooth segmentation allows for a more detailed evaluation of the volume and position of tooth eruption and root resorption (*Tanna, AlMuzaini & Mupparapu, 2021*). Tooth segmentation also serves as a valuable tool in guiding orthognathic surgeries by providing precise anatomical information (*Elnagar, Aronovich & Kusnoto, 2019*; *de Waard et al., 2022*). Additionally, in dental implantology (*Orentlicher, Horowitz & Abboud, 2012*) and transplantation (*Wu et al., 2019*), tooth segmentation contributes to accurate positioning and navigation during procedures, ultimately improving the overall success rates of these interventions.

The conventional approach to 3D tooth segmentation is manual and involves generating handcrafted outlines of teeth slice by slice. However, this method requires doctor experience and is highly subjective and time-consuming. Due to its drawbacks, the manual method is often used as a parameter for other segmentation methods (*Sercan, Barı̧s & Aysun, 2021*). With the evolution of computer vision technology, various semi or fully automatic image segmentation methods have been introduced. However, when applied to CBCT images, these algorithms need to be further improved to accommodate inherent image complexity. The challenges faced by these segmentation algorithms primarily include diverse tissue types on CBCT images and the subtle boundaries and structural intricacies of teeth (*Singh et al., 2020*; *Anwar et al., 2018*). Furthermore, variations in image quality and artifacts, such as noise and distortions, introduce uncertainties in segmentation tasks (*Singh et al., 2020*; *Anwar et al., 2018*). Moreover, real-time or near-real-time processing in clinical settings requires segmentation algorithms to be computationally efficient without compromising accuracy.

Scholars and researchers have responded to these challenges by proposing pragmatic segmentation algorithms aimed at elevating the quality of automatic 3D medical image segmentation. In tooth segmentation, semiautomatic methods are used for the initial epoch. These methods are primarily based on predefined mathematical models to assist operators in completing semisegmentation tasks, significantly reducing the time required for tooth segmentation. In recent years, propelled by the introduction of deep learning (DL), fully automatic methods have experienced notable advancements. These models, which are proficient at learning intricate image features, diminish the need for manual intervention and achieve efficient tooth segmentation.

Although numerous models have been proposed, there is a paucity of review articles on tooth segmentation algorithms. To address this gap, a recent study by *Polizzi et al. (2023)* extensively examined 23 research studies, scrutinizing the technological advancements in automatic tooth segmentation methods employing CBCT. Notably, the study predominantly concentrated on the application of DL methods for achieving fully automatic segmentation, providing limited coverage of semiautomatic approaches and overlooking insights from pivotal conference reports in the field. This article aims to fill this void by conducting a comprehensive review of semiautomatic or fully automatic tooth segmentation methods. Our objective is to carefully categorize and discuss the included articles, elucidating how they automate the process of tooth segmentation by applying and refining existing image segmentation algorithms. Moreover, we analyzed the results of these methods, demonstrating their continuous progress in reducing manual intervention and enhancing the accuracy and robustness of the models. This provides in-depth insights into the intricate advancements in this field.

## SURVEY METHODOLOGY

The literature search for this study was conducted in December 2023 utilizing electronic databases, namely, PubMed, Web of Science (WOS), and IEEE Xplore (IEEE). The search applied the following query parameters: (((tooth) OR (teeth)) AND ((segmentation) OR (segment))) AND (((dental) AND (CT)) OR (CBCT)). Except for the patent articles we found on the WOS, all the studies were included.

The primary stages of the literature search are illustrated in Fig. 1. After applying the queries to the aforementioned databases, we obtained a total of 1,140 articles, 419 from PubMed, 649 from WOS and 72 from IEEE. After eliminating duplicate articles by title comparison, 667 records were retained.

Then, we manually examined the titles and abstracts to remove articles that did not address tooth segmentation in CBCT images. After this phase, a total of 61 articles were retained. Finally, we conducted a thorough examination of the full text, excluding two review articles and four articles that were not focused on automating the segmentation process. Our study included a total of 55 articles on tooth segmentation.

## RESULTS

### Classification and yearly distribution of included articles

The articles included in our review primarily arose from the rapidly evolving landscape of fully automatic and semiautomatic image segmentation algorithms developed in recent years. The methods employed for tooth segmentation can be broadly categorized into two groups: knowledge-based methods and deep learning-based methods.

As illustrated in Fig. 2, studies conducted before 2019 predominantly focus on knowledge-based methodologies, whereas in the post-2019 era, most studies highlight deep learning-based approaches. Notably, the year 2022 had the highest number of relevant articles, underscoring the ongoing exploration on tooth segmentation.

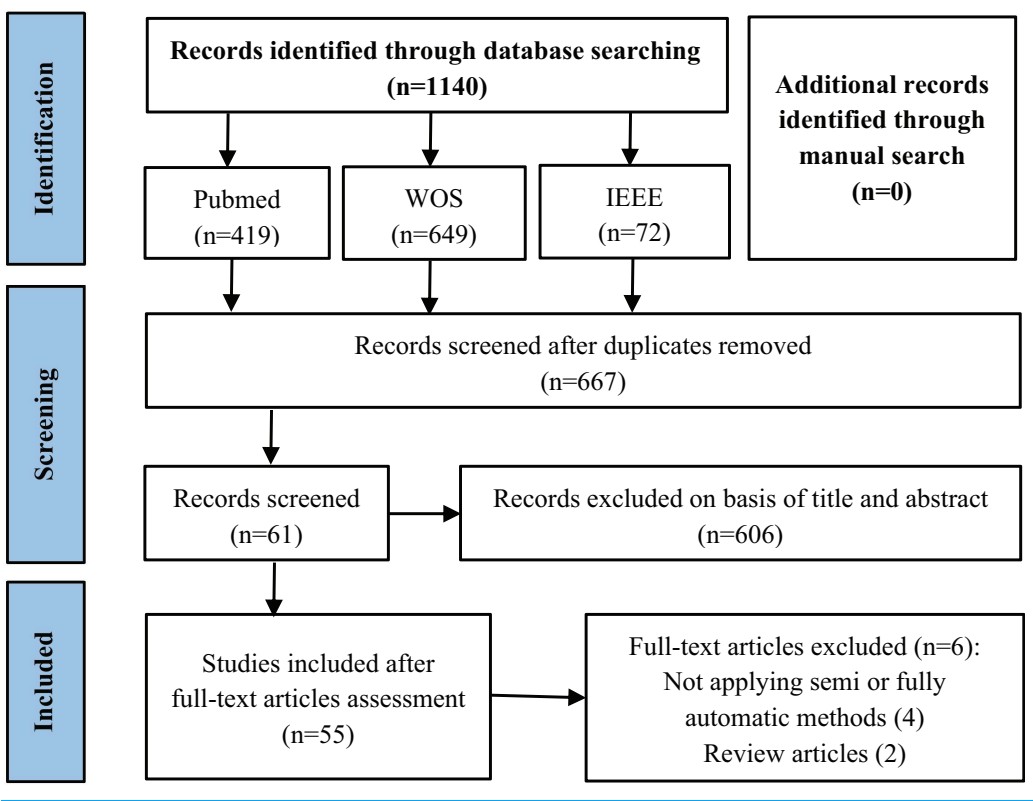

**Figure 1 Flowchart of the literature review article selection process.**

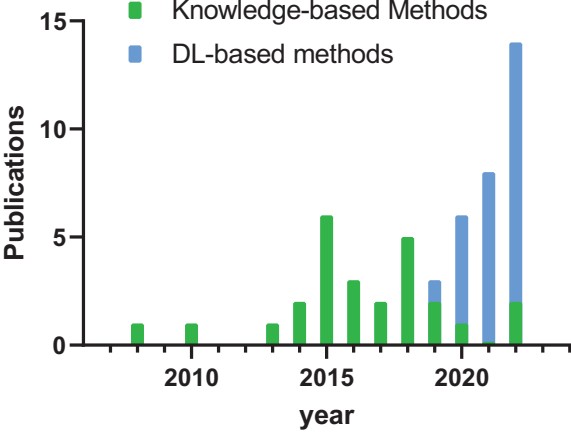

**Figure 2 Histogram of publication year of literature.**

## Evaluation metrics

Segmentation outcomes in various studies are evaluated by diverse assessment metrics, broadly categorized into three groups: overlap-based metrics, distance-based metrics, and volume-based metrics. The calculation methods for these metrics are detailed in Table 1.

Overlap-based metrics gauge the dissimilarity in the overlap between automated segmentation outcomes and manual segmentation outcomes, leveraging the confusion

**Table 1 Evaluation metrics employed to perform measurement of automatic tooth segmentation.**

| Metric | Abbreviation | Definition |
|---|---|---|
| **Overlap-based metrics** | | |
| Accuracy | Acc | $Acc = \dfrac{\lvert Y_r \cap Y_p \rvert + \lvert (1 - Y_r) \cap (1 - Y_p) \rvert}{\lvert U \rvert} = \dfrac{TP + TN}{TP + TN + FP + FN}$ |
| Precision | Pre | $Pre = \dfrac{\lvert Y_r \cap Y_p \rvert}{\lvert Y_p \rvert} = \dfrac{TP}{TP + FP}$ |
| Positive predictive value | PPV | $PPV = Pre$ |
| Recall | Rec | $Rec = \dfrac{\lvert Y_r \cap Y_p \rvert}{\lvert Y_r \rvert} = \dfrac{TP}{TP + FN}$ |
| Sensitivity | Sen | $Sen = Rec$ |
| Specificity | Spe | $Spe = \dfrac{\lvert (1 - Y_r) \cap (1 - Y_p) \rvert}{\lvert 1 - Y_r \rvert} = \dfrac{TN}{TN + FP}$ |
| Dice similarity coefficient | Dice | $Dice = \dfrac{2 \lvert Y_r \cap Y_p \rvert}{\lvert Y_r \rvert + \lvert Y_p \rvert} = \dfrac{2TP}{2TP + FP + FN}$ |
| Jaccard similarity coefficient | Jac | $Jac = \dfrac{\lvert Y_r \cap Y_p \rvert}{\lvert Y_r \cup Y_p \rvert} = \dfrac{TP}{TP + FP + FN}$ |
| Intersection over union | IoU | $IoU = Jac$ |
| **Distance-based metrics** | | |
| Average symmetric surface distance | ASSD | $ASSD(A, B) = \dfrac{d(A, B) + d(B, A)}{2}$ <br> $where \; d(A, B) = \dfrac{1}{N} \sum_{a \in A} \min_{b \in B} a - b$ |
| Maximum symmetric surface distance | MSSD | $MSSD(A, B) = \max(h(A, B), h(B, A))$ <br> $where \; h(A, B) = \max_{a \in A} \min_{b \in B} a - b$ |
| 95% Hausdorff distance | 95HD | $95HD = \max\left(h^{95\%}(A, B), h^{95\%}(B, A)\right)$ <br> $where \; h^{95\%}(A, B) = \max_{a \in A} \min_{b \in B^{95\%}} a - b$ |
| **Volume-based metrics** | | |
| Volume overlap error | VOE | $VOE = 1 - \dfrac{\lvert Y_r \cap Y_p \rvert}{\lvert Y_r \cup Y_p \rvert}$ |
| Relative volume difference | RVD | $RVD = \dfrac{\lvert Y_r - Y_p \rvert}{\lvert Y_r \rvert} \; or \; \dfrac{\lvert Y_p - Y_r \rvert}{\lvert Y_r \rvert}$ |

**Note:**
mm, millimeters; %, percents. Indicates the pixels in the reference standard (ground truth), and is the pixels in the automatic segmentation. |.| represents the number of voxels. ||.|| represents the L2 norm. a and b are corresponding points on the boundary of A and B.

matrix. This matrix, presented in tabular format, provides a comprehensive overview of a model's predictions compared to the actual outcomes. The model is divided into four quadrants, defining four distinct outcomes: true positive (TP), false positive (FP), false negative (FN), and true negative (TN). TP denotes the accurate segmentation of foreground pixels, FP indicates the count of pixels erroneously classified as foreground, FN represents the total count of pixels incorrectly classified as background, and TN signifies the correct classification of background pixels (*Qiu et al., 2021*). These elements facilitate the computation of various secondary indicators, such as accuracy (Ac), precision (Pre),

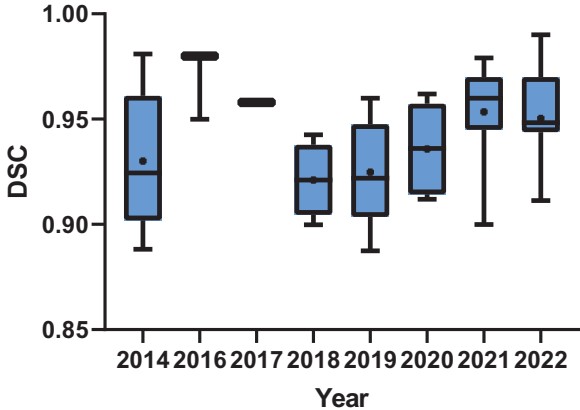

**Figure 3  Box plot of the dice score of the publications by year.**

recall (Rec), specificity (Spe), Dice similarity coefficient (Dice), and Jaccard similarity coefficient (Jac), or the intersection over union (IoU).

Distance-based metrics assess the geometric disparities between the segmentation result and the ground truth. In the context of tooth segmentation, commonly used distance-based metrics include the average symmetric surface distance (ASSD), maximum symmetric surface distance (MSSD), and 95% Hausdorff distance (95 HD).

Volume-based metrics concentrate on the volume and shape attributes of the segmented region, encompassing the volume overlap error (VOE) and relative volume difference (RVD). These three metric categories collectively provide an exhaustive evaluation, quantifying the performance of segmentation algorithms from diverse perspectives. These algorithms empower researchers to gain a comprehensive understanding of their accuracy and reliability in medical image analysis.

## Result analysis

In the realm of results analysis, Dice, ASSD, and MSSD emerged as the most frequently employed metrics in knowledge-based methodological research. Conversely, in deep learning-based methods, Dice coefficient and IoU/Jac are the most commonly utilized metrics. Nevertheless, due to the diverse range of evaluation metrics employed across various studies for assessing segmentation outcomes, it is challenging to quantitatively evaluate the results of all publications under a unified standard. This article conducts a brief statistical analysis of publications that adopt Dice coefficient as one of the metrics for assessing result accuracy. As depicted in Fig. 3, there was an upward trend in the Dice scores of publications over time, indicating the ongoing advancement in tooth segmentation methods.

The subsequent sections of this article provide a distinct summary and discussion of the included literature on various image segmentation methods. We believe that by elucidating the principles of different methods and presenting their application instances in the specific task of tooth segmentation, readers will gain a thorough understanding of the current status and developmental trajectories of this field.

**Table 2  Summary of the knowledge based methods.**

| Authors | Method | Performance | | | | | | | | | | Strengths | Weaknesses |
|---|---|---|---|---|---|---|---|---|---|---|---|---|---|
| | | Dice | ASSD (mm) | MSSD (mm) | RVD | PPV | Sen | Spe | Ac | IoU/Ja | 95% HD (mm) | | |
| Threshold based segmentation methods | | | | | | | | | | | | | |
| Marin et al. (2015) | Adaptive thresholds | / | / | / | / | / | / | / | / | / | / | Automatic delimitation for touching teeth | Limited capacity for pathology |
| Indraswari et al. (2018) | Histogram threshold, Region split and merge | / | / | / | / | / | 0.8022 | 0.9831 | 0.9775 | / | / | High accuracy and recall | Sensitivity improvement needed |
| Edge based segmentation methods | | | | | | | | | | | | | |
| Pavaloiu et al. (2015a, 2015b) | Canny operators | / | / | / | / | / | / | / | / | / | / | Reducing computational load | Potential challenges in fully automation |
| Region based segmentation methods | | | | | | | | | | | | | |
| Kang et al. (2015) | Region growing | / | / | / | 0.0229 ± 0.0056 | / | / | / | / | / | / | Fast and accurate | Requires user-defined seed points |
| Jiang et al. (2019) | Region growing | 0.9352 ± 0.0057 | 0.29 ± 0.04 | 2.17 ± 0.63 | / | / | / | / | / | / | / | Effective in multiple root tooth | Sharp edges in segmented curves |
| Naumovich, Naumovich & Goncharenko (2015) | Watershed algorithm | / | / | / | / | / | / | / | / | / | / | User-friendly software | Limited in CBCT spatial resolution |
| Galibourg et al. (2018) | Watershed algorithm | / | / | / | / | / | / | / | / | / | / | Clear geometric visualization | Manual refinement needed |
| Kakehbaraei, Seyedarabi & Zenouz (2018) | Watershed algorithm | / | / | / | / | 0.8681 | 0.9414 | 0.9994 | 0.9993 | / | / | Superior in sensitivity, specificity, and accuracy | Susceptibility to artifacts |
| Contour based segmentation methods | | | | | | | | | | | | | |
| Barone, Paoli & Razionale (2016) | B-spline | / | / | / | / | / | / | / | / | / | / | Enhanced clearness of tooth contours | Limited to anterior and premolar teeth with mono-radicular anatomies |
| Zhang et al. (2016) | Deformable surface | / | / | / | / | / | / | / | / | / | / | Effective in complex structures | Manual data integration |

(Continued)

| Authors | Method | Performance | | | | | | | | | | Strengths | Weaknesses |
|---|---|---|---|---|---|---|---|---|---|---|---|---|---|
| | | Dice | ASSD (mm) | MSSD (mm) | RVD | PPV | Sen | Spe | Ac | IoU/Ja | 95% HD (mm) | | |
| Harrison et al. (2019) | Deformable surface | 0.9219 ± 0.0231 | 0.24 ± 0.05 | / | / | / | / | / | / | 0.8559 ± 0.0389 | 1.54± 0.52 | Efficient surface evolution, high accuracy | Limited parameter details |
| Gao & Chae, 2010 | Variational level set | / | / | / | / | / | / | / | / | UCI: 0.987 2nd LM: 0.983 UM: 0.965 | / | Successful crown and root separation | Sensitivity to noise |
| Ji, Ong & Foong (2014) | Variational level set | 0.981 ± 0.008 | / | 1.13 ± 0.57 | / | / | / | / | / | 0.964 ± 0.011 | / | Enhanced segmentation accuracy | Manual shape prior input |
| Wang et al. (2018) | Variational level set | | / | / | 0.1141 | / | / | / | / | / | / | Improved RVD over existing methods | Manual selection of initial slice and center point |
| Gan et al. (2015) | Hybrid level set | Incisor: 0.89 ± 0.02 Canine: 0.92 ± 0.08 Premolar: 0.92 ± 0.02 Molar: 0.94 ± 0.01 | Incisor: 0.29 ± 0.03 Canine: 0.27 ± 0.02 Premolar: 0.29 ± 0.05 Molar: 0.3 ± 0.08 | Incisor: 1.25 ± 0.58 Canine: 1.06 ± 0.40 Premolar: 1.28 ± 0.72 Molar: 1.52 ± 0.75 | Incisor: 0.38 ± 0.13 Canine: 0.49 ± 0.09 Premolar: 0.38 ± 0.10 Molar: 0.52 ± 0.18 | / | / | / | / | / | / | Significant accuracy improvement | Open bite requirement, impacted teeth challenges |
| Gan et al. (2017) | Hybrid level set | / | / | / | / | / | / | / | / | / | / | Overcomes limitations of existing methods for angled teeth | Complexity in VOI estimation, reliance on accurate 3D axis |

| Authors | Method | Performance | | | | | | | | | | Strengths | Weaknesses |
|---|---|---|---|---|---|---|---|---|---|---|---|---|---|
| | | Dice | ASSD (mm) | MSSD (mm) | RVD | PPV | Sen | Spe | Ac | IoU/Ja | 95% HD (mm) | | |
| *Gan et al. (2018)* | Hybrid level set | Incisor: 0.90 ± 0.03 Canine: 0.92 ± 0.01 Premolar: 0.92 ± 0.02 Molar: 0.94 ± 0.01 | Incisor: 0.28 ± 0.03 Canine: 0.27 ± 0.01 Premolar: 0.28 ± 0.02 Molar: 0.3 ± 0.06 | Incisor: 1.21 ± 0.63 Canine: 0.92 ± 0.32 Premolar: 1.23 ± 0.54 Molar: 1.41 ± 0.66 | / | / | / | / | / | / | / | Low ASSD and MSSD | Sensitive to metal artifacts, struggle with angled teeth coherence |
| *Qian et al. (2021)* | Hybrid level set | / | 0.1789 ± 0.0552 | 0.8561 ± 0.1436 | / | / | / | / | / | / | / | Multimodal precision, automatic segmentation | Dependency on scanned data |
| *Jiang et al. (2022)* | Hybrid level set | 0.9407 ± 0.0218 | 0.16 ± 0.08 | / | / | / | / | / | / | / | 1.58± 1.17 | Excels in impacted or malposed teeth | Requires manual user initiation |
| **Methods based on specific theories** | | | | | | | | | | | | | |
| *Mortaheb, Rezaeian & Soltanian-Zadeh (2013)* | Mean shift algorithm | / | / | / | / | 0.8167 | 0.8536 | 0.9836 | 0.9733 | / | / | Outperforms other methods | Challenges in low-contrast root slice segmentation |
| *Mortaheb & Rezaeian (2016)* | Mean shift algorithm | / | / | / | / | 0.7277 | 0.8324 | 0.9835 | 0.9762 | / | / | Robust metal artifact reduction | Limited contrast in root slices |
| *Pei et al. (2016)* | Random walk | Anterior: 0.98 Premolar: 0.98 Molar: 0.95 | / | / | / | / | / | / | / | Anterior: 0.97 Premolar: 0.96 Molar: 0.89 | / | High IoU | Relies on exemplars |
| *Evain et al. (2017)* | Graph cut | 0.958 ± 0.023 | / | Anterior: 0.11 Premolar: 0.12 Molar: 0.26 | / | / | 0.972± 0.023 | 0.999 ± 0.001 | 0.998 ± 0.001 | / | / | Overcomes poor image quality, achieves precise results with reasonable computation time | Challenges with artifacts from metallic objects |
| *Kakehbaraei et al. (2023)* | Distance transform | / | / | / | / | / | / | / | / | / | / | Artifacts reduction | Manual adjustment of image artifacts |

## Knowledge-based methods

Primarily, knowledge-based approaches leverage existing image processing methods, mathematical modeling, and domain-specific expertise to recognize grayscale features in CBCT images, thereby achieving the automatic extraction of tooth structures from the background. These methods can be divided into distinct classes: threshold-based segmentation methods, edge-based segmentation methods, region-based segmentation methods, contour-based segmentation methods and other methods based on specific theories. Table 2 provides a summary of these knowledge-based approaches.

## Threshold-based segmentation methods

Threshold-based methods execute image segmentation by categorizing each pixel based on its gray value compared to a predefined threshold. These encompass fixed, histogram, iterative, and adaptive threshold models.

*Marin et al. (2015)* employ an adaptive threshold model to filter images and identify potential tooth edges. *Indraswari et al. (2018)* applied a histogram threshold model for grayscale binarization. However, in *Indraswari et al. (2018)*, due to the concentrated gray levels between teeth and alveolar bones, the histogram threshold model was employed in conjunction with other segmentation approaches to achieve a high accuracy of 0.9775.

## Edge-based segmentation methods

Edge-based segmentation methods identify disjointed regions in an image by discerning variations in grayscale, color, and texture among different objects. Detecting target image edges is central to accomplishing segmentation tasks. Prominent differential operators for edge detection include the Sobel, Roberts, Prewitt, Canny, and Laplacian operators.

The approach advocated by *Pavaloiu et al. (2015a, 2015b)* employs Canny operators for edge detection, leveraging anatomical knowledge of teeth to alleviate computational burdens and enhance outcomes. Therefore, as indicated by *Pavaloiu et al. (2015b)*, this approach has achieved an approximately sixfold increase in speed and an approximately 10% improvement in accuracy compared to their previous work.

## Region-based segmentation methods

Region-based methods are applied to segment teeth in CBCT images by identifying regions of homogeneity or similarity. These algorithms include seeded region growing, region split and merge and watershed algorithms. Compared to edge-based methods, these approaches place greater emphasis on the overall contextual information within the segmented regions, enabling them to adapt to gradual variations or blurred tooth boundaries. To validate the performance of these methods in different patients, several scholars (*Kang et al., 2015*; *Jiang et al., 2019*; *Kakehbaraei, Seyedarabi & Zenouz, 2018*) started collecting clinical CBCT images (7–10 patients) and conducted quantitative statistical analyses on the segmentation results.

### Region growing algorithm

The region growing algorithm starts from seed points in the tooth region, iteratively evaluating whether each pixel meets the predefined grayscale value. The process involves

incrementally incorporating neighboring pixels into the segmented region until no additional pixels meet the similarity conditions, thereby effectively demarcating the entirety of the tooth region.

Kang et al. (2015) propose a segmentation model based on seeded region growth combined with iterative methods, achieving a 2.4-fold speed improvement using multicore processors. Jiang et al. (2019) used an improved per-pixel region growing model combined with an edge detection model and achieved an average Dice coefficient of 0.9352 for multiple root teeth. However, the precision and reliability methods in Kang et al. (2015) and Jiang et al. (2019) hinge on the meticulous selection of seed points and the optimization of criteria.

### Region split and merge algorithm

The region split and merge algorithm involves recursively partitioning images based on homogeneity criteria and merging regions with analogous attributes. Segmentation is sequentially refined by considering pixel values and facilitating the identification of individual teeth.

Indraswari et al. (2018) applied the region merging algorithm before histogram thresholding was performed to distribute the grayscale intensity inside the teeth and alleviate oversegmentation and undersegmentation.

### Watershed algorithm

The watershed algorithm views CBCT images as topographic landscapes, where the gradients of grayscale values of different pixels are analogous to variations in terrain elevation. This method combines gradient information from the image with points that potentially represent tooth locations, creating a series of watershed lines. These lines serve to extract regions representing teeth from the background.

To compensate for the shortcomings of existing software products for low-contrast CBCT images, Naumovich, Naumovich & Goncharenko (2015) developed software and algorithms based on watershed transformation. The method proposed by Kakehbaraei, Seyedarabi & Zenouz (2018) utilizes the marker-controlled watershed (MCW) algorithm and achieves outstanding results in terms of sensitivity (0.9414), specificity (0.9994), and accuracy (0.9993). Galibourg et al. (2018) employed a watershed-based method to conduct automatic tooth segmentation *via* CBCT and compared its performance with that of a validated semiautomatic segmentation method. The results indicate a difference in volume between them, and the difference increases with increasing voxel size on CBCT.

Overall, the aforementioned methods have achieved precise delineation of tooth edges. However, since they all depend on the low-level visual properties of the image (such as grayscale and texture), continuous fine-tuning of parameters and manual intervention are needed. Consequently, more flexible framework models are needed to seamlessly integrate the intrinsic low-level visual attributes of CBCT images with higher-level attributes, such as prior knowledge of dental morphology, for more robust and automated segmentation.

## Contour-based segmentation methods

Contour-based methods for tooth segmentation refer to image processing techniques that primarily rely on identifying and delineating the boundaries or contours of teeth in dental images. These methods utilize mathematical curves, such as B-spline curves or active contours, to capture the intricacies and variable shapes of teeth.

### B-spline-based model

B-spline serves as a mathematical representation for creating smooth curves. The core principles of B-spline-based methods include defining a B-spline curve that closely aligns with the tooth contours, strategically positioning control points to shape the curve, and utilizing optimization to improve segmentation accuracy. The approach in *Barone, Paoli & Razionale (2016)* initially extracts 2D tooth contours in multiple planes and then automatically reconstructs the overall 3D tooth shape through B-spline algorithms. Since these planes are automatically defined based on specific anatomical morphologies, their method is similar to the approaches mentioned earlier (*Pavaloiu et al., 2015a*, *2015b*) and are discussed later (*Ji, Ong & Foong, 2014*; *Wang et al., 2018*), as they are influenced by anatomical considerations.

### Deformable surface model

The deformable surface model sets a method of tooth segmentation with shape transformations and adjustments. *Harrison et al. (2019)* and *Zhang et al. (2016)* proposed segmentation methods that apply deformable surface models. *Zhang et al. (2016)*, after testing on a collected set of 510 clinical CBCT images, verified that the proposed method has an average processing speed of approximately 3–6 s and a higher volume variation ratio. This indicates that the proposed method can converge more quickly than the other methods and requires fewer iterations to achieve better performance. *Harrison et al. (2019)* reported an average Dice score of 0.921 for teeth with irregular shapes. However, due to some constraints, the evolution of the surface toward distant parts of the tooth may be somewhat restricted, leading to occasional segmentation errors at the root tip (apex) or at incisal edges and cusp locations (*Harrison et al., 2019*).

### Active contour model

The active contour model (ACM) is also a type of curve evolution algorithm. The model first defines an initial contour of teeth in CBCT images and then adjusts the contour's shape by minimizing the energy function. This process aims to better adapt the contour to the boundaries of the teeth. As summarized in *Xu, Yezzi & Prince (2000)*, these methodologies are typically categorized into parametric and geometric ACMs. The parametric ACM, rooted in the Lagrange framework, constrains its energy function solely to selecting curve parameters, thereby imposing limitations on its broader applicability. In contrast, the geometric ACM relies on geometric measurements rather than curve expression parameters and is employed to address issues found in the parametric ACM.

The integration of the level set method with curve evolution theory has significantly facilitated the development of the geometric ACM and thereby expanded the applicability

of the ACM (*Chang, Li & Xu, 2013*). Consequently, most ACM-based methods are also referred to as level set-based methods.

### Single level set model

The single-level set model (SLSM) is a fundamental approach based on the framework of level set methods. *Gao & Chae (2008)* applied the SLSM for tooth root segmentation, addressing challenges posed by complex image conditions and the issue of root branching. However, when dealing with tooth crown segmentation, especially for adjacent tooth crowns (*Jiang et al., 2022*), the performance of SLSM may be limited by certain factors.

### Variational level set model

Compared to the SLSM, the variational level set model achieves greater flexibility by minimizing the comprehensive energy function. To avoid missing boundaries between the crowns of two adjacent teeth, in *Gao & Chae (2008, 2010)*, a coupled variational level set method was proposed to address the conflation between adjacent tooth regions. This method separates touching teeth by generating a virtual common boundary.

In contrast, prior shape information is introduced to guide the image segmentation algorithm with greater automation. This allows for automatic layer-by-layer segmentation after specifying a rough initial contour in a single slice, contributing to increased algorithmic efficiency (*Gao & Chae, 2010*) and addressing issues of leakage and shrinkage. *Ji, Ong & Foong (2014)* enhanced the variational level set framework and achieved an average Dice score of 0.981 by utilizing the thickness of the tooth dentine wall as a robust shape prior. Similarly, the variational level set model employed by *Wang et al. (2018)* leverages the initial intensity and shape information from the preceding slice to enhance and shape-constrain the subsequent slice.

### Hybrid level set model

Additionally, *Gan et al. (2015)* conducted extensive research and proposed a hybrid level set model. They employed a global convex level set model to extract the connected region of teeth and alveolar bones from CBCT images (*Gan et al., 2015, 2018*). Subsequently, individual teeth and alveolar bone are separated from the connected region. Moreover, to achieve a more precise segmentation of angled teeth, *Gan et al. (2017)* presented a protocol that initially extracted the region of interest (ROI) and the corresponding axis of the target tooth from CBCT images. Subsequently, local images of the target tooth are rotated to align its axis perpendicular to the transverse section. Ultimately, the hybrid level set model is employed to iteratively extract the tooth contours from the rotated images.

Furthermore, *Jiang et al. (2022)* alternately integrated the edge-based hybrid level set method with the geodesic active contour (GAC) method, controlling the evolution of these two models *via* a switch. Compared to several of the previously mentioned methods (*Gan et al., 2015, 2018*; *Harrison et al., 2019*; *Jiang et al., 2019*), this approach yields the smallest ASSD values (0.16 ± 0.08 mm).

In recent years, with advancements in dental image acquisition techniques, segmentation outcomes from level set methods can be combined with image data from other modalities to increase precision. *Qian et al. (2021)* used tooth crown meshes

obtained from laser scans to guide the tooth segmentation process automatically for corresponding CBCT data. They also replace the tooth crowns of the reconstructed meshes from CBCT data with those obtained from laser scans. The results obtained using multimodal data demonstrated a more accurate representation of teeth than models segmented from CBCT images.

## Other methods based on specific theories

In addition to the aforementioned methodologies, certain studies explore the application of specific machine learning or image segmentation algorithms to automate the tooth segmentation process.

### Mean shift algorithm

Derived from a clustering algorithm, the mean shift algorithm was applied to tooth segmentation by *Mortaheb, Rezaeian & Soltanian-Zadeh (2013)*. *Mortaheb & Rezaeian (2016)* introduce a multistep method that employs a least squares support vector machine (LSM) to mitigate metal artifacts and a mean shift algorithm for automatic segmentation. This method not only demonstrated high sensitivity and accuracy at that time but also achieved initial robust metal artifact reduction.

### Random walk model

The random walk model, a direct identification method utilizing random numbers to determine the search direction, is also applied in tooth segmentation. *Pei et al. (2016)* proposed a method that obtains the initial segmentation of teeth through a pure random walk approach. As an iterative refinement, they employ regularization through 3D exemplar registration and label propagation *via* random walks with soft constraints. This integrated approach, which combines semisupervised label propagation and regularization through 3D exemplar registration, allows for a more comprehensive utilization of information in the dataset.

### Graph cut

The minimum cut-max flow algorithm is used to partition the image into foreground and background parts. The semiautomatic framework proposed by *Evain et al. (2017)* acquires shape priors through a statistical shape model and graph cut optimization, perfecting the outcome through the application of morphological opening and the watershed algorithm.

### Distance transform model

The primary concept behind binary image distance transformation is to convert a binary image into a grayscale image by assessing the spatial distance between points (target and background points). *Kakehbaraei et al. (2023)* employed a distance transform model and succeeded in reducing artifacts through histogram adjustment and morphology filtering in axial slices.

## Summary

Typically, the knowledge-based methodologies explored in this study leverage prior knowledge and heuristic principles, demonstrating a notable level of interpretability in

addressing distinctive attributes of dental structures and segmentation outcomes. Nevertheless, since the effectiveness of these methods is heavily contingent upon the accuracy and comprehensiveness of prior knowledge, they are potentially less robust when confronted with complex variations in tooth structures or specific scenarios. Additionally, due to the significant degree of manual intervention, these methods may be susceptible to subjective influences.

Therefore, many researchers have introduced artificial intelligence (AI)-based techniques to achieve fully automatic tooth segmentation. Compared to manual operation and semiautomatic segmentation, these methods intelligently learn features of teeth in CBCT images through end-to-end training on large-scale data. Therefore, they can adapt to more diverse datasets and varied tooth morphologies.

## Deep learning-based methods

Machine learning (ML) is a subset of AI which addresses the question of how to build computers that improve automatically through experience (*Jordan & Mitchell, 2015*). As an advanced type of ML, deep learning (DL) employs artificial neural networks with multiple layers (deep neural networks) to model and process complex data representations. It discovers intricate structure in large data sets by using the backpropagation algorithm to indicate how a machine should change its internal parameters that are used to compute the representation in each layer from the representation in the previous layer (*LeCun, Bengio & Hinton, 2015*). Medical image segmentation primarily involves deep neural network architectures, including convolutional neural networks (CNNs), recurrent neural networks (RNNs), and encoder-decoder structures.

CNNs are among the most successful and widely utilized architectures in DL, particularly in computer vision tasks (*Minaee et al., 2022*). They employ convolutional, nonlinear, and pooling layers to extract features efficiently. Through locally connected units and shared weights within receptive fields, CNNs reduce parameter count while enabling hierarchical feature learning across multiple resolutions (*Aloysius & Geetha, 2017*).

RNNs are specifically designed for processing sequential data. However, they sometimes struggle with long sequences due to their inability to capture long-term dependencies and susceptibility to gradient vanishing or exploding problems (*Minaee et al., 2022*). LSTM, a specific type of RNN, addresses these issues by incorporating gates (input gate, output gate, and forget gate) to regulate information flow into and out of memory cells, allowing storage of values over arbitrary time intervals (*Byeon et al., 2015*).

Encoder-decoder structures are popular in sequence-to-sequence modeling for image-to-image translation, where the output can be an enhanced version of an image or a segmentation map (*Cho et al., 2014*).

These architectures constitute the fundamental elements of numerous state-of-the-art automatic image segmentation models based on DL. Notably, these models have gained extensive traction in the domain of dentistry, effectively mitigating limitations in knowledge-based tooth segmentation methodologies. Categorically, tooth segmentation

models based on DL can be broadly classified into semantic segmentation models and instance segmentation models.

## Semantic segmentation models

Semantic segmentation entails partitioning an image into distinct regions based on features such as grayscale, color, spatial texture, and geometric shape (*Mo et al., 2022*). Table 3 provides a summary of these semantic segmentation models. Consequently, these models specifically achieve pixel-level segmentation, extracting all teeth from both the upper and lower jaws as a unified entity from the background.

### CNN-based models

Semantic segmentation models for teeth are primarily based on CNNs. Occasionally, these models integrate other knowledge-based algorithms or DL networks to improve segmentation outcomes.

*Ma & Yang (2019)* integrated a lightweight CNN with a classical level set method known as the distance regularized GAC model, achieving state-of-the-art segmentation results with a Dice score of 0.8874 and a PPV of 0.9982. *Wang et al. (2021)* introduce an innovative mixed-scale dense (MS-D) CNN that incorporates scales within each layer and establishes dense connections among all feature maps. This design enhances feature extraction and transmission capabilities for tooth segmentation, resulting in an improvement of 0.945 in Dice score compared to that of *Ma & Yang (2019)*. Employ a CNN-based efficient neural network (Enet) model to reduce network parameters while maintaining model accuracy. The model was tested on the public tooth CT image dataset in West China, demonstrating its fast speed in tooth segmentation (one second per CBCT scan).

In general, these models emphasize the simplification and optimization of CNN architectures and parameters, resulting in enhanced network efficiency and accurate tooth segmentation (*Wang et al., 2021*; *Ma & Yang, 2019*).

### U-Net-based models

Proposed by *Ronneberger, Fischer & Brox (2015)*, U-Net has emerged as the predominant DL model for medical/biomedical image semantic segmentation due to its ability to extract contextual information efficiently with shorter training times and fewer sample data (*Siddique et al., 2021*). The U-Net model, primarily structured as an encoder-decoder model, draws inspiration from the full convolutional network (FCN, a CNN-based model characterized by exclusively incorporating convolutional layers) proposed by *Long, Shelhamer & Darrell (2015)*. It comprises a contraction path for context capture and a symmetric expansion path for precise positioning. The expansion path is more or less symmetrical with respect to the contraction path, resulting in a U-shaped architecture (*Ronneberger, Fischer & Brox, 2015*). *Hsu et al. (2022)* demonstrated that the performance of U-Nets for automatic tooth segmentation in CBCT images varies among different training strategies, such as 2D, 2.5D and 3D U-Nets. The studies included in this review were primarily based on models that use 2D and 3D U-Nets.

**Table 3 Summary of the semantic segmentation models.**

| Authors | No.of cases | Performance | | | | | | | | | Strengths | Weaknesses |
|---|---|---|---|---|---|---|---|---|---|---|---|---|
| | | Dice | ASSD (mm) | RVD | IoU/Ja | 95%HD (mm) | PPV | Sen | Spe | F1 | | |
| **CNN-based models** | | | | | | | | | | | | |
| Ma & Yang (2019) | 100 | 0.8874 | / | / | / | 1.987 | 0.9982 | / | / | / | State-of-the-art results in a challenge dataset | Limitations in touching teeth, highlight artifacts and small teeth |
| Wang et al. (2021) | 30 | 0.945 ± 0.021 | 0.204 ± 0.061 | / | / | / | / | / | / | / | Achieving high Dice score | Potential issues with thin structures and image quality |
| **2D U-Net based models** | | | | | | | | | | | | |
| Li et al. (2020) | 24 | 0.9526 | 0.1445 | / | 0.9142 | / | / | 0.9549 | 0.9577 | / | High segmentation accuracy | Challenges with deep tooth roots and partial loss. |
| Lee et al. (2020) | 102 | 0.912 | / | / | / | / | / | 0.881 | 0.945 | / | Increased average precision, effective adjacent tooth distinction | Requiring realignment for varying poses |
| Rao et al. (2020) | 110 | 0.9166 | 0.25 | / | / | / | / | / | / | / | Performance enhancement | Not mentioned |
| Yang et al. (2021) | 10 | 0.9791 ± 0.0145 | / | / | 0.9525 ± 0.0271 | / | 0.9733 ± 0.0159 | / | / | 0.9824 ± 0.0324 | Superior tooth segmentation performance | Complex reliance on mathematical procedures |
| Wu et al. (2022) | 100 | / | / | / | 0.8637 | / | / | / | / | / | Robust, precise and effective | Limited dataset, challenges in complex dental occlusion |
| Tao & Wang (2022) | 500 | / | / | / | 0.8373 | / | 0.8591 | / | / | / | Better segmentation performance and efficiency | Further improvement of segmentation performance needed |
| **3D U-Net based models** | | | | | | | | | | | | |
| Liu et al. (2021) | 170 | / | / | / | / | / | / | / | / | / | Swift, precise dental segmentation | Limited tooth landmark detection in the coarse stage |
| Deng et al. (2023) | 61 | Upper: 0.97 Lower: 0.96 | Upper: 0.1 ± 0.1 Lower: 0.01 ± 0.01 | / | / | / | / | / | / | / | Fast, efficient automatic tooth segmentation with high precision and recall | Limited assessment of segmentation in the presence of artifacts |
| Dot et al. (2022) | 453 | Upper: 0.95 ± 0.02 Lower: 0.94 ± 0.02 | / | / | / | / | / | / | / | / | Peformace surpassing 2DU-Net | Minor noise and shadow artifacts |
| **V-Net based models** | | | | | | | | | | | | |
| Chen et al. (2020) | 25 | 0.936 ± 0.012 | 0.363 ± 0.145 | 0.088 ± 0.031 | 0.881 ± 0.019 | / | / | / | / | / | Suitable for diverse teeth, non-open bite positions | Potential blurring of tooth boundaries |
| Dou et al. (2022) | 40 | 0.952 | 0.15 | / | 0.902 | 2.12 | 0.996 | / | / | / | Emphasizing tooth geometry, detail, and multiscale features | Room for improvement in detail preservation |

*2D U-Net*

2D U-Net utilizes a 2D image as the fundamental unit for input data, while 2Da, 2Dc, and 2D U-Net employ axial, coronal, and sagittal slices, respectively, as input data units (*Hsu et al., 2022*).

*Rao et al. (2020)* and *Lee et al. (2020)* substituted specific convolutional layers within the network and achieved Dice scores exceeding 0.915. *Rao et al. (2020)* substituted regular convolutional layers in 2D U-Net with three specialized deep bottleneck architectures (DBAs). This effectively deepens the network, optimizes feature extraction and enhances accuracy in localizing the tooth. *Lee et al. (2020)* replaced certain convolutional layers with dense blocks and introduced spatial dropout layers between contraction and extraction paths. The proposed UDS-Net (U-Net + dense block + spatial dropout) achieves improved segmentation performance with a reduced parameter count compared to the original U-Net.

The expanded modules were introduced by *Wu et al. (2022)* and *Tao & Wang (2022)* into the 2D U-Net. *Wu et al. (2022)* integrate a local feature enhancement module (LE) into the decoder network to fully leverage accurate semantic and location context information across the input image. *Tao & Wang (2022)* introduce an attention module into the 2D U-Net network to amplify the importance of critical information. These models (*Wu et al., 2022*; *Tao & Wang, 2022*) optimize the 2D U-Net-based model, and their IoU scores are approximately 0.84.

Moreover, the 2D U-Net can be synergistically combined with other DL-based or knowledge-based models to mitigate some of its inherent limitations. *Li et al. (2020)* integrate a 2D attention U-Net with a bidirectional convolution-LSTM (BDC-LSTM). This approach compensates for the 2D U-Net's sole extraction of intraslice contexts by enabling the BDC-LSTM to extract interlayer information from the tooth root sequence. *Yang et al. (2021)* amalgamate the 2D U-Net model with an enhanced ACM, addressing the challenges that DL methods may face in automatic topology changes. Compared to other 2D U-Net-based models (*Rao et al., 2020*; *Lee et al., 2020*; *Wu et al., 2022*; *Tao & Wang, 2022*), these two approaches can achieve higher Dice/IoU scores with a smaller dataset (10 in *Yang et al. (2021)* and 24 in *Li et al. (2020)*).

*3D U-Net*

The structure of the 3D U-Net closely resembles that of the 2D U-Net, with the distinction that 2D operations are substituted with 3D operations. Employing a cuboid as the unit for input data, the 3D U-Net can process the entire 3D image instead of individual slices during training (*Çiçek et al., 2016*). *Liu et al. (2021)* introduced a coarse-to-fine 3D U-Net-based framework, named SkullEngine, which was designed for high-resolution segmentation and large-scale landmark detection in the skull. *Deng et al. (2023)* studied 61 patients from the digital archive of the Department of Oral and Maxillofacial Surgery at Houston Methodist Hospital between January 2021 and December 2021 and independently validated the segmentation accuracy of SkullEngine for teeth (Dice score for upper jaw teeth: 0.97; Dice score for lower jaw teeth: 0.96).

Additionally, *Dot et al. (2022)* assessed the performance of another segmentation model named nnU-Net, an out-of-the-box tool proposed by *Isensee et al. (2021)*. nnU-Net generates three U-Net configurations: a 2D U-Net, a full-resolution 3D U-Net, and a cascaded 3D U-Net (*Isensee et al., 2021*). In comparison with those of *Deng et al. (2023)*, the patients included in their study by *Dot et al. (2022)* presented diverse anatomical deformities and had undergone orthognathic surgery. *Dot et al. (2022)* demonstrated that nnU-Net performs well on CBCT images of such patients (Dice score for upper jaw teeth: 0.95; Dice score for lower jaw teeth: 0.94).

### V-Net-based models

A volumetric FCN model named V-Net is derived from 3D U-Net, attaining enhanced segmentation efficiency through the incorporation of residual architectures in each convolutional stage (*Milletari, Navab & Ahmadi, 2016*). *Chen et al. (2020)* applied a modified V-Net architecture to handle tooth regions and tooth surface prediction simultaneously. *Dou et al. (2022)* integrated an attention mechanism and a self-regulatory mechanism into the V-Net network structure, achieving higher Dice (0.952) and IoU (0.902) scores and lower ASSD (0.15 mm) values than did the methods proposed by *Chen et al. (2020)*.

However, the semantic segmentation approaches mentioned above lack the ability to perform more detailed analysis and manipulation for each individual tooth. Therefore, many studies introduce instance segmentation models for individual tooth segmentation.

## Instance segmentation models

Instance segmentation entails discriminating and outlining individual teeth in CBCT images as a distinct instance (*Tian et al., 2022*). This approach surpasses semantic segmentation by offering a nuanced comprehension of the spatial boundaries and type of each tooth. Table 4 shows a summary of these instance segmentation models. These models can be broadly classified into two groups: two-stage models and one-stage models.

### Two-stage models

Within the domain of two-stage instance segmentation models, research has focused primarily on two paths: bottom-up semantic segment-based models and top-down detection-based models.

#### Bottom-up semantic segment-based models

Bottom-up semantic segment-based models generate instance masks by initially classifying pixels through semantic segmentation and subsequently distinguishing different instances of the same kind through clustering or other metric learning methods.

*Cui, Li & Wang (2019)* introduced a two-stage model named ToothNet for automatic tooth instance segmentation in CBCT images. They initiate the process by extracting the tooth edge with the help of a network that comprises a single encoder, nine convolutional layers and three decoder branches. Subsequently, a network constructed from the 3D region proposal network (RPN) is applied to generate region proposals for further tooth

**Table 4 Summary of the instance segmentation models.**

| Authors | No.of cases | Performance | | | | | | | | Strengths | Weaknesses |
|---|---|---|---|---|---|---|---|---|---|---|---|
| | | Dice | ASSD (mm) | RVD | IoU/Ja | 95% HD (mm) | PPV | Sen | Spe | | |
| Bottom-up two-stage models | | | | | | | | | | | |
| *Cui, Li & Wang (2019)* | 20 | 0.9198 | / | / | / | / | 0.9775 | / | / | Superior results wtith efficient training | Sensitive to extreme gray values |
| Anchor-based top-down two-stage models | | | | | | | | | | | |
| *Chung et al. (2020)* | 175 | / | 0.15 ± 0.04 | / | / | 0.86 ± 0.44 | / | 0.93 ± 0.04 | 0.93 ± 0.07 | High sensitivity and recall | Computational concerns |
| *Lahoud et al. (2021)* | 314 | / | 0.10 ± 0.06 | / | 0.87 ± 0.03 | / | / | / | / | AI matches manual accuracy | Limited to single/ double-rooted teeth, potential sensitivity to artifacts |
| *Lee et al. (2022)* | 120 | / | / | / | 0.704 | / | / | 0.932 | 0.919 | Surpassing existing approaches, especially in challenging scenarios | Not mentioned |
| Point-based top-down two-stage models | | | | | | | | | | | |
| *Wu et al. (2020)* | 20 | 0.962 | 0.122 | / | / | / | / | / | / | Effective metal artifact handling | Limited by dataset scale |
| *Duan et al. (2021)* | 20 | ST: 0.96 ± 0.01 <br> MT: 0.96 ± 0.00 | ST: 0.10 ± 0.02 <br> MT: 0.14 ± 0.02 | ST: 0.05 ± 0.02 <br> MT: 0.05 ± 0.01 | / | / | / | / | / | Addressing complex tooth morphology | Challenges in full-mouth CBCT |
| *Cui et al. (2021)* | 100 | 94.8 ± 0.4 | 0.18 ± 0.02 | / | / | 1.52 ± 0.28 9 | / | / | / | Achieving state-of-the-art results | Not mentioned |
| *Cui et al. (2022)* | 4938 | 0.9254 | 0.21 | / | / | / | / | / | 0.921 | High accuracy, efficiency improvement | Limited tooth crown surface detail |
| One-stage models | | | | | | | | | | | |
| *Chung et al. (2020)* | 175 | / | 0.15 ± 0.04 | / | / | 0.86 ± 0.44 | / | 0.93 ± 0.04 | 0.93 ± 0.07 | Effective overcoming of metal artifacts | Not mentioned |
| *Shaheen et al. (2021)* | 186 | 0.90 ± 0.03 | / | / | 0.82 ± 0.05 | 0.56 ± 0.38 | / | 0.98 ± 0.02 | 0.83 ± 0.05 | High precision, recall; fast processing | Potential segmentation quality degradation |
| *Fontenele et al. (2022)* | 175 | Anterior: 0.95 ± 0.03 <br> Premolars: 0.97 ± 0.03 <br> Molars: 0.97 ± 0.03 | / | / | Anterior: 0.91 ± 0.05 <br> Premolars: 0.94 ± 0.06 <br> Molars: 0.95 ± 0.04 | Anterior: 0.25 ± 0.34 <br> Premolars: 0.17 ± 0.38 <br> Molars: 0.19 ± 0.43 | Anterior: 0.99 ± 0.01 <br> Premolars: 0.99 ± 0.01 <br> Molars: 0.99 ± 0.01 | Anterior: 1.00 <br> Premolars: 1.00 <br> Molars: 1.00 | Anterior: 0.91 ± 0.05 <br> Premolars: 0.94 ± 0.05 <br> Molars: 0.94 ± 0.04 | Robust, accurate, time-efficient, even with dental fillings | Observing minor impact of fillings |

| Authors | No.of cases | Performance | | | | | | | | Strengths | Weaknesses |
|---|---|---|---|---|---|---|---|---|---|---|---|
| | | Dice | ASSD (mm) | RVD | IoU/Ja | 95% HD (mm) | PPV | Sen | Spe | | |
| Jang et al. (2022) | 97 | 0.9441 ± 0.0310 | 0.27 ± 0.12 | / | / | 1.34 ± 0.86 | / | 0.9618 ± 0.0467 | 0.9317 ± 0.0583 | Robust to metal artifacts | Limited to single/ double-rooted teeth, dependence on panoramic images |
| Ayidh Alqahtani et al. (2023) | 215 | 0.99 ± 0.06 | / | / | 0.99 ± 0.02 | 0.12 ± 0.15 | 0.99 ± 0.01 | 0.99 ± 0.02 | 0.99 ± 0.01 | Outperforms existing methods | Limited to specific CBCT device |
| Li et al. (2022) | 350 | 0.9113 ± 0.0045 | / | / | 0.8480 ± 0.0057 | 1.00 ± 0.27 | / | 0.9213 ± 0.0029 | 0.9123 ± 0.0089 | Superior performance, especially in challenging scenarios | Not mentioned |
| Xie, Yang & Chen (2023) | 10 | 0.9480 ± 0.0257 | / | / | 0.9023 ± 0.0437 | / | / | 0.9484 ± 0.0287 | / | Superior accuracy and stability | Occasional interference-related failures |
| Gerhardt et al. (2022) | 170 | / | / | / | 0.97 | 0.15 | / | / | / | Achieving high accuracy | Potential false positives |

identification. This model is the first DL solution for tooth instance segmentation *via* CBCT.

However, the bottom-up approach relies primarily on unextracted low-level features during the first step of segmentation. Therefore, in *Cui, Li & Wang (2019)*, the Dice score for segmentation on the test set was approximately 0.91, leaving room for further improvement.

*Top-down detection-based methods*

The top-down detection methods initiate instance region detection in the image, followed by pixel-level segmentation of the candidate area. Initially, the location or boundaries of the instance are determined through target detection; these methods acquire the ROI of each individual tooth and subsequently conduct semantic segmentation in the ROIs.

In algorithms for ROI detection, a bounding box serves as a rectangular representation delineating the spatial coordinates of a detected object in the image (*Padilla, Netto & Da Silva, 2020*). The anchor-based object detection method depends on the generation of predefined anchors, which are subsequently refined by predicted bounding box offsets to localize the tooth ROIs precisely in the CBCT images.

*Chung et al. (2020)* implemented a faster R-CNN (*Ren et al., 2017*) framework based on the RPN for initial ROI detection. *Lahoud et al. (2021)* utilized the RPN with a feature pyramid network (FPN) (*Lin et al., 2017*) model for precise single-tooth localization and bounding box determination. *Lee et al. (2022)* employed a 3D FCN layer followed by an encoding-decoding structure based on a 3D hourglass network (*Xu & Takano, 2021*) to

extract the bounding box. These methods enhance the ROI detection accuracy, with average precision at an IoU of 50 (AP50) of approximately 90% and object inclusion ratio (OIR) values exceeding 99.9%.

Instead of bounding boxes, point-based object detection models apply key points for the precise localization of ROIs. Its distinct advantage over anchor-based detection lies in the absence of anchor boxes, which simplifies the complex hyperparameter choices in the network required to generate a large number of proposed regions (*Duan et al., 2021*).

*Wu et al. (2020)* introduced a two-level method featuring a center-sensitive heatmap mask strategy at the global stage for precise tooth center determination and a DenseASPP-U-Net at the local stage for detailed segmentation of individual teeth. The approach proposed by *Duan et al. (2021)* is also grounded in the differentiation of individual teeth through heatmaps and box regressions.

The method proposed by *Cui et al. (2021)* and *Cui et al. (2022)* is designed to extract the centroid and skeleton of each tooth as coarse-level morphological representations. Its robustness and generalizability were evaluated and validated on the largest dataset (a total of 4,938 CBCT scans) to date (*Cui et al., 2022*).

In general, the majority of two-stage instance segmentation models use top-down approaches. These methods exhibit lower dependence on computing power and rely extensively on precise target detection.

### One-stage models

Inspired by single-stage target detection, which omits the independent and explicit extraction of candidate regions, current one-stage segmentation methods are based on the FCN framework and demonstrate enhanced segmentation accuracy and computational efficiency. For instance, *Xie, Yang & Chen (2023)* employ a fully convolutional one-stage object detector (FCOS) model proposed by *Tian et al. (2019)* to identify each individual tooth. The average Dice score of the model is 0.9480, with a standard deviation of 0.0257, indicating the outstanding and stable performance of the proposed model.

Essentially, instance segmentation can be conceptualized as region-level, instance location-aware semantic segmentation, where the same pixel may carry different semantics in distinct regions. *Jang et al. (2022)* proposed a three-step method involving the generation of 2D panoramic images from 3D CT images and the identification and segmentation of individual teeth in 2D panoramic images. Additionally, *Shaheen et al. (2021)* and *Gerhardt et al. (2022)* both employed 3D U-Net-based models, leveraging rough segmentation results for ROI identification. Furthermore, *Li et al. (2022)* initially applied a 3D U-Net-based model to segment four quadrants of teeth from input CBCT data as ROIs before employing a semantic graph attention mechanism (SGANet) based on graph convolutional networks (GCNs) proposed by *Defferrard, Bresson & Vandergheynst (2016)* for tooth segmentation in each quadrant. SGANet explicitly incorporates anatomical topology information to learn fine-grained discriminative features, reducing confusion in delineating boundaries between adjacent teeth (*Li et al., 2022*).

To enhance the robustness and accuracy of segmentation models across diverse CBCT datasets, various methodologies have been devised to mitigate the impact of metal artifacts,

dental filling materials, and orthodontic brackets. The method proposed by *Fontenele et al. (2022)* comprises multiple configured 3D U-Nets, which progress step by step from rough detection to refined segmentation of teeth with high-density dental filling material. Similarly, *Ayidh Alqahtani et al. (2023)* validated a segmentation pipeline configured with multiple 2D U-Nets. These methods demonstrate high classification and segmentation accuracy (over 0.99) for teeth filled with high-density dental filling material (*Fontenele et al., 2022*) or with brackets (*Ayidh Alqahtani et al., 2023*).

## Summary

The outlined deep learning-based approaches overcome the limitations of traditional methods and achieve fully automatic and robust tooth segmentation. The significant advantages of DL in tooth segmentation include precise segmentation, flexibility in handling various imaging challenges, and continuous advancements in model architectures. Collectively, these aspects establish DL as a pivotal tool in the evolution of dental image segmentation methodologies.

Finally, when different DL models are used for automated tooth segmentation tasks, dentists should take into consideration patient needs, possible side effects during dental practice, and the characteristics of the radiographic image dataset.

## DISCUSSION

### Challenges and advances

Automating tooth segmentation encounters numerous challenges, such as the intricacy of dental and bone tissues, issues arising from low-quality images, and difficulties in procuring sufficient training datasets.

Research indicates that segmentation inaccuracies predominantly manifest as fine anatomical structures such as cervical margins, pits and fissures (*Galibourg et al., 2018*) or apical lesions in the context of multiple-rooted teeth (*Qian et al., 2021*). The presence of surrounding periodontal tissues often obscures the true boundaries of teeth (*Mortaheb, Rezaeian & Soltanian-Zadeh, 2013*; *Ji, Ong & Foong, 2014*; *Li et al., 2020*; *Xie, Yang & Chen, 2023*), thereby intensifying segmentation complexities. Segmenting extremely small teeth (*Ma & Yang, 2019*), irregularly arranged teeth (*Gan et al., 2017*), and impacted third molars (*Qian et al., 2021*) with diverse eruption positions and shapes (*Wu et al., 2020*) is challenging. Additionally, the proximity of adjacent teeth (*Mortaheb, Rezaeian & Soltanian-Zadeh, 2013*; *Kakehbaraei, Seyedarabi & Zenouz, 2018*; *Kakehbaraei, Seyedarabi & Zenouz, 2018*; *Ma & Yang, 2019*) or closely positioned teeth during occlusion (*Gan et al., 2017*; *Kakehbaraei, Seyedarabi & Zenouz, 2018*; *Qian et al., 2021*; *Wu et al., 2020*) may induce mutual interference, leading to segmentation failure in delineating joint contours.

Several quality parameters of CBCT images, such as low resolution, noise, contrast deficiency (*Galibourg et al., 2018*; *Mortaheb, Rezaeian & Soltanian-Zadeh, 2013*), and the presence of metal artifacts (*Kang et al., 2015*; *Gan et al., 2017*; *Kakehbaraei, Seyedarabi & Zenouz, 2018*), negatively impact segmentation outcomes. In response to these constraints, segmentation pipelines undergo continual refinement. During the preprocessing phase, images typically require homogenization and enhancement before automated software-

driven segmentation to address artifacts and enhance data consistency (*Kakehbaraei, Seyedarabi & Zenouz, 2018*; *Galibourg et al., 2018*; *Naumovich, Naumovich & Goncharenko, 2015*). Several studies project 3D images into 2D (*Galibourg et al., 2018*) or regress tooth poses and realign them (*Chung et al., 2020*; *Lee et al., 2022*) to mitigate the spatial complexities of CBCT images, thereby contributing to improved segmentation model performance.

In DL-based tools, the efficacy of automatic segmentation models is occasionally constrained by the scale and annotation accuracy of the training dataset. *Cui et al. (2022)* emphasized the significance of large-scale, multicenter, and real-clinical data over synthesized small-sized datasets. *Ayidh Alqahtani et al. (2023)* also noted that a large labeled training dataset is essential to avoid overfitting a model, enhance its learning and optimization, and effectively capture the inherent data distributions. Recently, specific data augmentation techniques, such as flipping, rotation, random deformation, and conditional generative models, have been implemented to augment the training dataset (*Cui et al., 2022*).

## Clinical application and ethical responsibilities of AI

The ongoing progress in automatic tooth segmentation technology has significantly streamlined the workflow in clinical digital dentistry, providing substantial benefits for patients and dental professionals. In comparison to traditional manual methods, automatic segmentation technology takes approximately 20–30 s to segment a CBCT scan, whereas manual segmentation requires approximately 3–5 h (*Wang et al., 2021*). Moreover, even in the presence of metal or other artifacts in CBCT images, the models of *Fontenele et al. (2022)* and *Ayidh Alqahtani et al. (2023)* can accurately perform segmentation without relying on additional image processing steps.

However, both dentists and computer programmers engaged in AI should recognize the ethical responsibilities associated with the application of AI in dentistry. This approach is particularly pertinent to issues of patient data privacy, compliance, and underlying errors in AI-driven methodologies. *Silva & Soto (2022)* emphasize that in the realm of AI-related research and clinical experiments, direct and candid communication with patients to obtain their consent is established prior to data collection. *Roy (2022)* and *Harvey & Gowda (2021)* stressed the importance of encryption and access restriction protocols to ensure the confidentiality of imaging data from clinical patients. They believe that this relies heavily on stringent cybersecurity measures, comprehensive regulatory frameworks, and regular auditing mechanisms to mitigate the risk of data breaches. Finally, acknowledging the possible errors in AI-driven approaches, *Pesapane et al. (2018)* suggested that dental practitioners must actively identify discrepancies while applying AI technology. Researchers, in turn, should conduct regular assessments and update AI systems based on clinical feedback. In summary, these considerations collectively contribute to ethical responsibility in the application and development of AI in dental practices.

## Future directions

First, future research endeavors should focus on the continuous optimization of tooth segmentation algorithms to adapt to increasingly diverse datasets and mitigate the impact of factors such as image artifacts and anatomical complexities. During the algorithm optimization process, researchers may contemplate the incorporation of techniques such as adaptive learning (*Kerr, 2016*) and transfer learning (*Weiss, Khoshgoftaar & Wang, 2016*) to enhance the algorithm's performance across various data distributions. Furthermore, for distinct types of dental structures, the use of advanced network architectures such as transformers can be considered (*Strudel et al., 2021*; *Dosovitskiy et al., 2020*). Exploring methods that leverage multiscale and contextual information is essential for capturing both detailed anatomical structures and their holistic relationships (*Strudel et al., 2021*). This integration may entail models with a larger number of parameters and more complex structures, necessitating increased computational resources and training time (*Dosovitskiy et al., 2020*).

Second, future research endeavors should focus on achieving precise segmentation of each internal structure while segmenting the entire tooth. This includes detailed delineation of the tooth crown and roots and accurate segmentation of structures such as enamel, dentin, dental bone, and root canals. By integrating advanced computer vision and DL technologies, researchers can strive to develop more refined and accurate segmentation models, thereby providing clinicians with more detailed information about dental structures.

Third, for future research on tooth segmentation techniques, segmentation techniques should be integrated with other advanced image processing technologies for a more comprehensive understanding and presentation of dental anatomical structures. For instance, combining 3D imaging (*Karatas & Toy, 2014*) and virtual reality (*Anthes et al., 2016*) technologies can offer dental professionals a more innovative 3D display of dental structures, leading to more precise surgical planning and preoperative predictions.

Generally, future research on tooth segmentation should further advance algorithm optimization, detailed structure segmentation, and the integration of multiple technologies to meet increasingly complex clinical demands. This approach will enhance the practicality and accuracy of segmentation techniques in dental practice.

## CONCLUSIONS

Fast and precise tooth segmentation in CBCT images constitutes a crucial component of digital dentistry. The automation of this intricate and time-consuming process poses a formidable challenge due to the intricate nature of oral tissues and variations in image quality. This research involved a comprehensive examination of 55 articles and presented an exhaustive analysis of fully automatic or semiautomatic tooth segmentation methods proposed over the past 15 years. Knowledge-based tooth segmentation approaches, encompassing threshold-based, edge-based, region-based, and contour-based methods, utilize existing mathematical models and achieve relatively accurate segmentation in specific test images. However, these methods have certain limitations when dealing with variable segmentation samples and often require varying degrees of manual intervention.

Current studies have shifted toward modern AI-driven approaches. Within the domain of DL-based tooth segmentation, approaches leveraging the U-Net model stand out due to their robust performance in medical image tasks. Instance segmentation models extend the ability to recognize and delineate individual teeth in CBCT images, offering a nuanced understanding of spatial boundaries and tooth types and contributing to more in-depth and detailed analysis. DL-based tooth segmentation significantly contributes to dental diagnostics and treatment planning. Prospective directions underscore the importance of algorithmic refinement, intricate structure segmentation, and the introduction of cutting-edge imaging technologies to effectively address ever-increasing clinical demands.

## ACKNOWLEDGEMENTS

A generative AI language model (ChatGPT-3.5; OpenAI, San Francisco, Calif) was used for language polishing and translation of a portion of the content.

### Funding

This work was supported by the Zhejiang Provincial Natural Science Foundation of China (LY18H140001), the National Natural Science Foundation of China (81400511), the Key R&D Program of Zhejiang (2023C03072), and the R&D Program of the Stomatology Hospital of Zhejiang University School of Medicine (RD2022DLYB03, RD2022JCEL04). The funders had no role in study design, data collection and analysis, decision to publish, or preparation of the manuscript.

### Grant Disclosures

The following grant information was disclosed by the authors:
Zhejiang Provincial Natural Science Foundation of China: LY18H140001.
National Natural Science Foundation of China: 81400511.
Key R&D Program of Zhejiang: 2023C03072.
Zhejiang University School of Medicine: RD2022DLYB03, RD2022JCEL04.

### Competing Interests

The authors declare that they have no competing interests.

### Author Contributions

- Qianhan Zheng conceived and designed the experiments, performed the experiments, analyzed the data, prepared figures and/or tables, authored or reviewed drafts of the article, and approved the final draft.
- Yu Gao conceived and designed the experiments, performed the experiments, prepared figures and/or tables, and approved the final draft.
- Mengqi Zhou conceived and designed the experiments, performed the experiments, prepared figures and/or tables, and approved the final draft.
- Huimin Li conceived and designed the experiments, prepared figures and/or tables, and approved the final draft.

- Jiaqi Lin conceived and designed the experiments, prepared figures and/or tables, and approved the final draft.
- Weifang Zhang conceived and designed the experiments, authored or reviewed drafts of the article, and approved the final draft.
- Xuepeng Chen conceived and designed the experiments, authored or reviewed drafts of the article, and approved the final draft.

## Data Availability

This is a literature review.

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
