# Peer review of "Semi or fully automatic tooth segmentation in CBCT images: a review"

_PeerJ Computer Science, doi:10.7717/peerj-cs.1994_

## Round 0.1 · original submission · Major Revisions

Dear authors,
You are advised to critically respond to all comments point by point when preparing a new version of the manuscript and while preparing for the rebuttal letter. Please address all the comments/suggestions provided by the reviewers.

Kind regards,
PCoelho

**Language Note:** PeerJ staff have identified that the English language needs to be improved. When you prepare your next revision, please either (i) have a colleague who is proficient in English and familiar with the subject matter review your manuscript, or (ii) contact a professional editing service to review your manuscript. PeerJ can provide language editing services - you can contact us at copyediting@peerj.com for pricing (be sure to provide your manuscript number and title). – PeerJ Staff

·

Basic reporting

Please consider breaking down the introduction section into smaller paragraphs to focus on specific aspects: the evolution of dental imaging methods, segmentation challenges, advancements in techniques, and the purpose of the review article.

Please enhance the citations by attributing direct sources to specific claims or findings. Also, emphasize the primary objective or novelty of the review earlier to engage readers and provide a clear direction for the manuscript.

To improve readability, please simplify complex terms and concepts, especially in explaining evaluation metrics. Please conclude this section by summarizing the research gap or indicating how the review will address it, serving as a transition to the subsequent sections.

Experimental design

Please organize the study design section into clearer subsections or headers to distinguish between traditional and deep learning-based approaches or different segmentation methodologies.

Please ensure explanations of methods are concise and structured, providing a brief overview before diving into specific studies or examples.

Please enhance credibility by attributing statements or findings to specific sources through in-text citations.

Consider providing a comparative analysis or summary table(s) after discussing each category of segmentation methods to aid readers in understanding the strengths, weaknesses, and notable findings of each approach.

Please include transitional sentences or paragraphs between different subsections to improve the flow and ensure a smooth transition between topics.

Please conclude this section with a brief summary, emphasizing the need for deep learning-based approaches in tooth segmentation and their potential benefits.

Validity of the findings

Overall, the study provides an extensive overview of tooth segmentation methodologies, covering both traditional knowledge-based methods and modern deep learning-based approaches. The detailed explanations of each methodology and the inclusion of citations contribute to the depth of understanding.

Comparative analyses or case studies would further enrich the discussion, providing a clearer understanding of the practical implications and limitations of these methods in dental imaging.

Areas for Improvement for Traditional Knowledge-Based Methods:

-Organizing the subsections under traditional methods with clear headings for better readability.
Incorporating more concise explanations, especially in lengthy sections, to enhance clarity.
-Providing examples or case studies to demonstrate the practical implications and limitations of each method.
-Offering a brief comparative analysis to highlight strengths and weaknesses of these approaches in tooth segmentation.

Areas for Improvement for Deep Learning-Based Methods:

-The section could benefit from clearer subsection divisions to distinguish between different types of deep learning models.
-Explaining complex methodologies with simpler language to make it more accessible for readers less familiar with technical jargon.
-Integrating a comparative analysis between different deep learning models to highlight their strengths and limitations in tooth segmentation.

Additional comments

1-) Please emphasize the critical role of tooth segmentation in digital dentistry workflows and its impact on clinical analysis and treatment planning.

2-) Please briefly compare the limitations and advantages of traditional methods against newer deep learning-based approaches to showcase the shift towards AI-driven methodologies.

3-) Please discuss the real-world implementation of advancements in tooth segmentation, emphasizing practical implications and benefits for dental professionals and patients.

4-) Please reiterate the potential for continued innovation in tooth segmentation, emphasizing the need for ongoing research to address remaining challenges and speculate on future advancements.

5-) Please acknowledge ethical responsibilities associated with AI-driven tools in dentistry, particularly concerning patient data privacy and regulatory compliance.

Reviewer 2 ·

Basic reporting

1. The abstract's first sentence is ambiguous. Please clarify the relationship between 3D reconstruction and tooth segmentation from CBCT images.
2. English writing is not good for different meanings.
3. The summarized chapters or paragraphs are too wide without concrete contents and their names are not clear.
4. Future directions lack evidence without support.

Experimental design

Study Design:

1. While the manuscript reviews semi or fully automatic tooth segmentation methods, the title only mentions "automatic tooth segmentation." Consider revising the title for accuracy.

2. Introduce a section titled "Modern/State-of-the-Art Knowledge-Based Methods" to complement the existing section on "Traditional Knowledge-Based Methods".

3. The "Traditional Knowledge-Based Methods" section does not cover B-spline methods for tooth segmentation. Please incorporate this relevant method into the discussion.

4. Rename the subsection "The Area-Based Methods" to better reflect its contents (the region growing, region split and merge and watershed algorithms), perhaps as "The Morphology-Based Methods" or choose a similar title.

5. The review overlooks significant articles, including:
- "Gao, Hui, and Oksam Chae. 'Touching tooth segmentation from CT image sequences using coupled level set method.' 2008 5th International Conference on Visual Information Engineering (VIE 2008). IET, 2008."
- "Gao, Hui, and Oksam Chae. 'Individual tooth segmentation from CT images using level set method with shape and intensity prior.' Pattern Recognition 43.7 (2010): 2406-2417."
- "Cui, Z. et al. (2021). Hierarchical Morphology-Guided Tooth Instance Segmentation from CBCT Images. In: Feragen, A., Sommer, S., Schnabel, J., Nielsen, M. (eds) Information Processing in Medical Imaging. IPMI 2021. Lecture Notes in Computer Science(), vol 12729. Springer, Cham. https://doi.org/10.1007/978-3-030-78191-0_12";

Validity of the findings

1. Lines 426 and 427 suggest that automatic tooth segmentation models will simplify digital dentistry workflows in the future. Provide examples or evidence demonstrating how these models will achieve such simplification.

General Comments:

The manuscript reviews the state-of-the-art semi or fully automatic tooth segmentation methods. However, the manuscript's language lacks clarity and professionalism. The article title and some section titles need adjustment, and crucial articles have been overlooked. Additionally, the "Challenges and Future Directions" section lacks supporting evidence for the proposed future directions.

Additional comments

No

Cite this review as

---

## Round 0.2 · Minor Revisions

Dear authors,

You are advised to check once more the details reported by the reviewers, and respond to all comments point by point when preparing a new version of the manuscript and while preparing for the rebuttal letter. Please address all the comments/suggestions provided by the reviewers. Make sure to proofread all the document against AI-generated sentences

Kind regards,
PCoelho

Reviewer 2 ·

Basic reporting

The manuscript is a review article. The title of the manuscript is very appropriate, and the content of the article meets the requirements of the journal. The introduction section of the article is well-written, with good English expression.
The section of “Traditional knowledge-based methods” summarizes various tooth segmentation methods clearly, but there are deficiencies in the writing of section or subsection titles. The title "Active contour model (ACM)" at line 288 should be changed to "Active contour model" and abbreviations should not be introduced at that position. The title "Single-level set model" at line 302 should be changed to "Single level set model" and unnecessary connectors should not be introduced.
There are several deficiencies in the section “Modern deep learning-based methods”. The use of the adjective "Modern" in the section title is not appropriate. The first paragraph about machine learning is not very clear, and it is highly suspicious that it was generated by ChatGPT, as similar descriptions can be generated in ChatGPT with "machine learning" as input. In such cases, ChatGPT's output is: "Machine learning is a subset of artificial intelligence (AI) that focuses on the development of algorithms and statistical models that enable computers to perform tasks without being explicitly programmed for those tasks". The second paragraph about DL has similar issues and does not explain the abbreviation "DL. The third and fourth paragraphs have similar issues.

Experimental design

This is a review article reviewing the articles and techniques for tooth segmentation. It is telling the right story.

Validity of the findings

Proof by the provided articles.

Additional comments

1. There are still grammar errors such as line 515 -> integrated. It is suggested to review grammars of this article.
2. Some sentence Line 50-51 state MRIs provide high-resolution image with less radiation exposure than traditional CT. It looks not correct statement since MRIs has no X-ray involved and so no radiation at all.

Cite this review as

·

Basic reporting

The manuscript is written in clear, professional English, provides an adequate introduction to the topic, and contextualizes the research within the field. The literature is well-referenced and relevant, indicating a broad and cross-disciplinary interest that aligns with the journal's scope. The introduction effectively introduces the subject, indicating the audience and motivation behind the review. The survey methodology is detailed, ensuring replicability, and the article is structured logically into coherent sections, discussing various tooth segmentation methods, their advantages, limitations, and future directions. This structured approach, combined with clear definitions and detailed exploration of current methodologies, suggests that the manuscript meets the professional article structure requirements, including figures and tables to support the findings. The review's broad coverage, without focusing solely on deep learning methods and including manual and semi-automatic approaches, provides a comprehensive overview of the field. Additionally, it identifies gaps in the literature, offering a unique perspective not recently reviewed, thus justifying the need for this review.

Experimental design

The Survey Methodology section of the manuscript provides a clear and detailed description of the methods used to perform the literature review, indicating a rigorous attempt to ensure comprehensive and unbiased coverage of the subject. The methodology includes criteria for inclusion and exclusion of studies, databases searched, and the search terms used, which are critical for replicability and assessing the scope of the review.

Sources are cited throughout the document, indicating a diligent effort to attribute ideas and findings to their original authors. The use of both direct quotations and paraphrasing appears appropriate, with citations provided in each case, supporting the credibility and ethical standards of the review.

The review is organized into logical sections that flow coherently from one to the next. These include the introduction, methodology, discussion of various tooth segmentation techniques, and conclusions. This structure facilitates easy navigation through the document and allows readers to understand the scope, findings, and implications of the review effectively.

Validity of the findings

The Conclusion section of the manuscript effectively links back to the original research question and is limited to the results and discussions presented within the document. It provides a succinct summary of the findings from the comprehensive examination of 55 articles over the past 15 years, focusing on fully automatic or semiautomatic tooth segmentation methods. The manuscript outlines the evolution from traditional knowledge-based methods to modern AI-driven approaches, specifically highlighting the significance of deep learning (DL)-based methods and their robust performance in medical image tasks, including tooth segmentation.

The argument developed throughout the manuscript is well-supported and logically structured, meeting the goals set out in the Introduction. The authors successfully bridge the gap between the current state of tooth segmentation techniques and the potential for future improvements. They emphasize the importance of algorithmic refinement, detailed structure segmentation, and the integration of cutting-edge imaging technologies to meet clinical demands effectively.

Furthermore, the Conclusion identifies unresolved questions and future directions, including the continuous optimization of tooth segmentation algorithms, achieving precise segmentation of each internal structure while segmenting the entire tooth, and integrating tooth segmentation techniques with advanced image processing technologies for a comprehensive understanding of dental anatomical structures. This forward-looking perspective not only addresses the goals introduced at the beginning but also opens avenues for future research, highlighting the manuscript's contribution to the field.

---

## Round 0.3 · accepted · Accept

Dear authors, we are pleased to verify that you meet the reviewer's valuable feedback to improve your research.

Thank you for considering PeerJ Computer Science and submitting your work.